# Application of Amorphous and Nanocrystalline Soft Magnetic Materials in Balanced-Force-Type Electromagnetic Relay

**DOI:** 10.3390/mi15030368

**Published:** 2024-03-08

**Authors:** Ding Ding, Jiaxin You, Xiangqian Cui, Yutong Xue, Xu Tan, Guofu Zhai

**Affiliations:** 1Institute of Reliability in Electrical Apparatus and Electronics, Harbin Institute of Technology, Harbin 150001, China; 22b906004@stu.hit.edu.cn (D.D.); 22s106116@stu.hit.edu.cn (X.C.); xueyutong@hityg.com.cn (Y.X.); gfzhai@hit.edu.cn (G.Z.); 2Shaanxi Qunli Electrician Co., Ltd., Baoji 721300, China; victorfm.tx@gmail.com

**Keywords:** electromagnetic relay, amorphous magnetic material, nanocrystalline magnetic material, dynamic characteristics

## Abstract

The magnetic properties of soft magnetic materials, including their saturation magnetic induction strength and permeability, significantly affect the dynamic characteristics of electromagnetic relays. However, the soft materials most commonly used for relays in the magnetic conductive components of electromagnetic systems, such as electrical pure iron, limit further relay design improvement and optimization to a certain extent. Thus, this paper proposes the use of amorphous and nanocrystalline soft magnetic materials with good high-frequency magnetic properties in magnetic circuits. A wavelet analysis was conducted on the high-frequency components of the coil current while the relay operated, and the corresponding magnetic materials were selected. Considering the challenges in processing amorphous and nanocrystalline materials and collecting test data for the accuracy verification of simulation methods, we prepared a scaled-up prototype for use in dynamic characteristic tests. The simulation method was improved, yielding more accurate simulation results regarding the relay’s dynamic characteristics. On this basis, six replacement schemes using amorphous and nanocrystalline materials were considered. The test results proved that this application could improve the relay’s dynamic characteristics. Finally, a full-size sample with an iron core consisting of nanocrystalline alloy 1K107B was prepared, and the conclusions were verified in tests.

## 1. Introduction

The magnetic properties of soft magnetic materials, the main magnetic circuit components of relays, affect the dynamic and static characteristics of the electromagnetic system and, in turn, the performance of the whole relay. Currently, the most common soft magnetic materials used for electromagnetic relays, such as soft iron, have good magnetic properties under DC or low-frequency conditions. However, when the relay operates or is released, high-frequency signals occur in the circuit due to the short switching time, which limits the performance of traditional soft magnetic materials.

Previous studies have not widely considered the properties of magnetic conductive component materials as an important factor affecting the dynamic characteristics of relays. Firstly, the currently used materials, such as DT4C and DT4E, appear to satisfy the requirements. Secondly, more attention has been paid to the air gap, as it is the main source of magnetoresistance in the magnetic circuit of the relay electromagnetic system [1].

In existing finite element simulation analyses of relay electromagnetic systems, the soft magnetic materials’ properties are usually depicted with only one magnetization curve [2]. However, there is a high-frequency component in the coil current when the electromagnetic relay operates, leading to a deviation between the currents in the simulation of the relay’s dynamic characteristics and the actual relay working process. However, this deviation is not obvious due to the poor performance of soft magnetic materials at high frequencies, indirectly causing the simulation results to meet the accuracy requirements. If the high-frequency component of the coil current is regarded as an important factor in the relay’s dynamic characteristics, it is necessary to introduce soft magnetic materials with good magnetic properties at high frequencies and compare them with traditional materials under the same working conditions. Accordingly, the existing simulation analysis methods that do consider the changes in magnetic properties may prove disadvantageous [3,4,5].

In recent years, the application of amorphous and nanocrystalline soft magnetic materials in electromagnetic devices has attracted increasing attention. Amorphous magnetic materials, e.g., metallic glass, are often formed by rapidly cooling molten liquids, vapor deposition, electrochemical deposition, high-energy ion implantation, and mechanical alloying. With uniform isotropic structures but no grain boundaries, amorphous magnetic materials have excellent properties not found in traditional crystalline materials [6], such as high strength and hardness, a low elastic modulus, corrosion resistance, and wear resistance. Some ferro-based amorphous materials also have good soft magnetic properties and have been widely used in transformers [7,8], motors [9,10], and other power electronic devices. For example, Guo, L. et al. improved the resolution of a MEMS-based micro-fluxgate sensor by using a magnetic core composed of a Fe-Co-B amorphous strip composite material [11]. Hsu, C et al. designed a micro-alternator using a shell coated with a ferro-based amorphous alloy soft magnetic material, reducing the noise characteristics by 15 dB compared with an uncoated shell [12]. Nanocrystalline soft magnetic materials often have a nanometer grain size, with high initial permeability and low coercivity in their spatial structure, which can be obtained by applying a certain heat treatment process to an amorphous matrix. Nanocrystalline materials outperform ordinary crystalline materials due to the size effect, surface effect, quantum size effect, and quantum tunneling effect of nano-scale grains. Several nanocrystalline alloy series, such as FeSiB, FeMB, and FeMC, have been developed [13] by considering the many excellent properties of ferro-based and cobalt-based amorphous alloys as soft magnetic materials. In addition, nanocrystalline materials have a higher saturation magnetic induction intensity and can replace cobalt-based amorphous alloys, crystalline permalloys, and ferrite [14], demonstrating wide application prospects in high-frequency power electronics equipment, such as common-mode inductors [15,16] and medium- and high-frequency transformers [17,18]. Yao, A. et al. used nanocrystalline materials for the stator and rotor cores of permanent magnet synchronous motors to reduce core loss [19]. Xiong, M. et al. replaced ferrite with a nanocrystal material in the core of a wireless power transfer (WPT) system to improve the load capacity [20]. Their many excellent magnetic properties, such as high saturation magnetic induction, high permeability, and low iron loss, have led to amorphous and nanocrystalline magnetic materials being widely used in high-frequency electronic devices. For this reason, these materials are worth considering for applications in relays [21,22,23].

This paper presents a new design idea for electromagnetic relays, with amorphous and nanocrystalline materials serving as soft magnetic parts in the magnetic circuit, including the iron core, yoke, and armature. Focusing on a cubic inch balanced-force-type sealed electromagnetic relay, widely used in aerospace, this study aims to obtain better dynamic characteristics in the relay by relying on the good high-frequency magnetic properties of nanocrystalline soft magnetic materials. The experimental procedure is detailed in Figure 1. Section 2 presents the soft magnetic material selection method and the corresponding cases. Section 3 establishes and tests a balanced force electromagnetic relay prototype based on the selected nanocrystalline magnetic materials. Section 4 details the simulation verification and subsequent analysis and comparison of the calculation results of different material replacement schemes. Finally, six replacement schemes of amorphous and nanocrystalline soft magnetic materials are provided and verified in Section 5, and the best scheme is selected for the preparation, testing, and analysis of a full-size nanocrystalline material relay.

## 2. Magnetic Material Selection

### 2.1. Wavelet Analysis Method

In this section, suitable nanocrystalline soft magnetic materials are screened through a wavelet analysis. When analyzing signals with traditional Fourier transform-based processing methods, only the frequency domain information of the signal is retained after the transform while time domain information is excluded, but it is equally important in some cases. Many novel processing and analysis methods have been proposed in recent years, such as the short-time Fourier transform and wavelet transform, which can simultaneously characterize signal information in the time–frequency domain.

As wavelet analysis can alter the shape of the time–frequency analysis window according to the frequency while the total area of the analysis window remains fixed, the wavelet analysis results have a higher frequency resolution but lower time resolution at low frequencies. At high frequencies, the opposite occurs. Therefore, wavelet analysis is suitable for the transient high-frequency signals displayed in normal signals.

The wavelet transform encompasses continuous and discrete forms. The function space in wavelet analysis is $L^{2}\left(R\right)$, and we have
(1)$$f\left(t\right)\in L^{2}\left(R\right)↔\int_{R}{\left|f\left(t\right)\right|}^{2}dt<+\infty$$

If $\psi \left(t\right)\in L^{2}\left(R\right)$, its Fourier transform $\hat{\psi }\left(\omega \right)$ satisfies the admissibility condition
(2)$$C_{\psi }=\int_{-\infty }^{+\infty }{\left|\omega \right|}^{-1}{\left|\hat{\psi }\left(\omega \right)\right|}^{2}d\omega <\infty$$

Here, ψ can be considered the basis wavelet. By stretching and shifting the basis wavelet, a wavelet sequence can be obtained:(3)$$\psi_{a,b}\left(t\right)={\left|a\right|}^{-1/2}\psi \left(\frac{x-b}{a}\right)$$
where a,b∈R and a≠0, *a* is the stretch factor, and *b* is the translation factor. The following defines the continuous wavelet transform (CWT) of basic wavelet ψ:(4)$$\left(W_{\psi }f\right)\left(a,b\right)=\langle f,\psi_{a,b}\rangle ={\left|a\right|}^{-1/2}\int_{-\infty }^{+\infty }f\left(t\right)\bar{\psi (\frac{t-b}{a})}dt$$

The wavelet transform converts the original one-dimensional signal into a two-dimensional one to analyze its time–frequency characteristics.

*a* and *b* are continuous in CWT, and the calculation of a continuous integral is generally time-consuming, creating inconvenience in digital signal processing. Discretized *a* and *b* values are favorable for the practical application of wavelet analysis. In practical engineering, this discretization is usually adopted for the discrete wavelet transform (DWT) of the original signal. Usually, we set
(5)$$a=a_{0}^{m},b=nb_{0}a_{0}^{m},m,n\in Z$$

By substituting it into the wavelet sequence, we have
(6)$$\psi_{m,n}\left(t\right)={\left|a_{0}\right|}^{-m/2}\psi \left(a_{0}^{-m}t-nb_{0}\right),m,n\in Z$$

The corresponding DWT is
(7)$$\left(W_{\psi }f\right)\left(a,b\right)=\langle f,\psi_{a,b}\rangle ={\left|a\right|}^{-1/2}\int_{-\infty }^{+\infty }f\left(t\right)\bar{\psi (a_{0}^{-m}t-nb_{0})}dt$$

In particular, when $a_{0}=2$ and $b_{0}=1$, the binary wavelet can be derived:(8)$$\psi_{m,n}\left(t\right)=2^{-m/2}\psi \left(2^{-m}t-n\right),m,n\in Z$$

### 2.2. Coil Current Wavelet Analysis

The sym3 function is selected as the wavelet function in this paper, and DWT and CWT are used to analyze the signal. The analysis results are presented in Figure 2.

As the relay is activated and operated, the signal displays a high-frequency component, which is consistent with the results of the discrete wavelet analysis. In the low-frequency part at the bottom, the corresponding component shows a large amplitude due to the overall upward trend of the signal. The local amplification of the wavelet time–frequency graph is shown in Figure 3.

Obvious high-frequency components can be observed as the relay is activated (period A) and operated (period B), consistent with the discrete wavelet analysis results. Considering these results, the amplitudes of the high-frequency components in periods A and B are roughly equal. Thus, the current signal may contain an 8 kHz component as the relay is activated and a 6.8 kHz component as the relay is operated.

### 2.3. Nanocrystalline Material Selection

Suitable magnetic materials are screened based on the wavelet analysis results. Considering that the applicable frequency of the magnetic material must include the high-frequency component of the coil current, the 1K101 amorphous alloy (0–10 kHz) and 1K107B nanocrystalline alloy (0–50 kHz) are chosen as the magnetic circuit replacement materials for the subsequent analysis and research.

The hysteresis loops of electrical soft iron DT4C, amorphous magnetic material 1K101, and nanocrystalline magnetic material 1K107B are presented as examples. As shown in Figure 4, the BH curve of DT4C gradually becomes rectangular as the frequency increases, which limits the coil current increase rate to some extent.

Figure 4 shows the BH curves of the DT4C, 1K101 amorphous alloy, and 1K107B nanocrystalline alloy soft magnetic materials at the frequency of 1–10 kHz. In the kHz frequency range, the rectangular shape of the pure iron DT4C’s BH curve gradually intensifies with the increase in frequency. Meanwhile, its coercivity and remanence are much higher than those of the amorphous and nanocrystalline materials. When the coil current contains more high-frequency components as the relay is activated and operated, the slope of DT4C’s BH curve is larger. As the current changes, the magnetic induction intensity changes ΔB are larger under equal magnetic field intensity increments ΔH, indicating that the back electromotive force is larger. As a result, the current increase is slower than that of the amorphous and nanocrystalline materials. Therefore, when replacing some components or the entire soft magnetic circuit with amorphous and nanocrystalline materials, the slope of the coil current waveform increases theoretically as the relay is activated and operated, thus shortening the period from activation to operation and improving the dynamic characteristics of the relay.

## 3. Prototype and Testing

Usually, a new design idea or method is subjected to theoretical analysis and preliminary evaluation and verification based on simulation; then, complete prototyping and testing are performed to further determine its feasibility and to ensure the accuracy of the simulation results. However, the existing relay simulation method only considers the magnetization curve under the DC condition of the material, which is not fully applicable to the research in this paper. Even if the simulation results are obtained, we cannot determine their accuracy. Therefore, we modify the initial concept by first machining a prototype for testing, improving the simulation method, verifying the accuracy of the method according to the test data, and finally completing the follow-up work based on the improved method.

### 3.1. Scaled-Up Model Design

Considering the difficulties and costs of machining amorphous and nanocrystalline materials into full-size (1 cubic inch in volume) relay components, we first design and trial a scaled-up relay model for the preliminary verification of the dynamic performance optimization effect of the proposed design and determine whether to process the full-size relay with amorphous and nanocrystalline materials.

As shown in Figure 5 and Figure 6, the electromagnetic system of the actual relay is scaled up four times, and the soft magnetic parts in the magnetic circuit are prepared with pure iron. Tests on the static and dynamic characteristics are performed on the model. Then, the core in the model is replaced with one composed of the nanocrystalline material, and the static and dynamic characteristics are tested again for comparison with the results of the original model. Figure 7 shows the finished product. The coil in the model has 994 turns, and its measured resistance is approximately 9.89 Ω.

### 3.2. Static Characteristic Test

The electromagnetic force–displacement curves of the scaled-up model are obtained using a JXF-II force–displacement curve tester, as shown in Figure 8.

The electromagnetic force curve presents the relationship between the force and displacement of the force sensor, while the simulation results present the relationship between the torque and rotation angle of the armature. Thus, the conversion relationship between the displacement and rotation angle must be clarified to facilitate the subsequent simulation of the relay’s dynamic characteristics. Figure 9 presents the conversion relationship.

*L* is the distance from the test point of the force sensor to the center of the rotating shaft, which is also the arm of force. With the armature at its initial position, $h_{0}$ is the vertical distance from the test point to the center of the rotating shaft, and the angle between the arm of force and the horizontal line is
(9)$$\alpha_{0}=\arcsin\left(\frac{h_{0}}{L}\right)$$

When the armature turns a certain angle from its initial position, we assume that the angle between the arm of force and the horizontal line is α, and the vertical distance between the test point of the force sensor and the center of the rotating shaft increases by Δh. Thus, the displacement of the sensor is Δh, and we have
(10)$$\alpha =\arcsin(\frac{h_{0}+\Delta h}{L})$$

At the end of the operation process,
(11)$$\alpha_{max}=\arcsin\left(\frac{h_{0}+\Delta h_{max}}{L}\right)$$

Thus, the force variation data (F−Δh) with displacement from the test system are processed as follows: (12)M=FLcosα
(13)$$\Delta \alpha =\alpha_{max}-\alpha =\alpha_{max}-\arcsin\left(\frac{h_{0}+\Delta h}{L}\right)$$

The curves (M−Δα) for the original model without material replacement can be obtained from the measured curves (F−Δh), as shown in Figure 10.

Then, the tests are repeated with the core in the original model replaced by one composed of the 1K107B nanocrystalline material, as shown in Figure 11.

The static characteristics of the model with replacement under voltages of 2.5 V and 3.5 V are compared with those of the original model, as shown in Figure 12.

### 3.3. Dynamic Characteristic Test

By testing the original scaled-up model, the pick-up voltage can be obtained as 1.8 V. According to the general relationship between the pick-up voltage, the rated voltage, and the power limit of the coil, the power voltage is set between 2.5 V and 3.5 V with a step of 0.5 V, thus obtaining the coil current and contact current curves, as shown in Figure 13.

The dynamic characteristics of the model with replacement are tested, and the results are as shown in Figure 14. The operation time before and after material replacement under different voltages is presented in Table 1.

The comparison of the dynamic characteristics of the model before and after material replacement (Figure 15) suggests that the operation time of the model with replacement is reduced by 6.5% to 8% and the current rising speed is increased near the beginning of the activation process and at the current rising stage upon the completion of the pick-up process, indicating that the material replacement improves the dynamic characteristics of the relay to a certain extent.

## 4. Simulation and Verification

### 4.1. Static Characteristic Simulation

The electromagnetic system model of the relay is built in Maxwell. Figure 16 shows the finite element model and meshed model.

According to the measured coil parameters described in Section 2, when a 1 V voltage is loaded on the coil, the current in the steady state is 0.1011 A and the coil’s ampere-turn value is 100.5 A. Under the excitation voltage *u*, the coil’s ampere-turn value is 100.5u. Therefore, the coil section is selected as the excitation current loading surface, the excitation is set to 100.5u ampere-turns as the input parameterized excitation, and the excitation type is stranded (hinge type).

The materials for each part of the electromagnetic system are defined, among which the solution domain is set as a vacuum, the permanent magnet material is defined as NdFeB35, the remanence is set to 1.1 T, and the coercivity is set to −890 kA/m, according to the parameters provided by the manufacturer, the coil is defined as copper, and the other armature, yoke, and core materials are defined as DT4C pure iron.

The CPU of the PC used is i7-8700. The maximum number of iterations is set to 10, the iteration error is 1%, and the mesh refinement is 30% in each iteration. We set the number of CPU parallel computing cores to 10. According to statistics, in the static process, the number of nodes in each position of the model fluctuates around 40,000, and the calculation time for each point is approximately 2 min.

Taking the original model as an example, the simulated curve under different voltages is obtained and compared with the curve M−Δα obtained by using the measured F−Δh after transformation, as shown in Figure 17. Under the coil voltage of 2.0 V, the simulated torque has a large error compared to the measured torque because the coil voltage is close to the pull-in voltage, and the static simulation model does not consider the influence of gravity. When the armature rotation angle is large, the influence of gravity is small, and the simulation and measurement results agree well. Under other coil voltages, the overall trend remains unchanged, and the numerical error is significantly reduced compared to the test results, indicating that the simulation results are accurate and can reflect the actual static characteristics of the scaled-up model.

In the model with material replacement, the material attributes of the iron core in the original simulation model are modified to match those of 1K107B to obtain the corresponding static characteristic curve, which is compared with the actual measurement, as shown in Figure 18. The simulated static characteristics of the replaced model are in good agreement with the actual measurements.

### 4.2. Dynamic Characteristic Simulation

The coil current waveform of the original model is investigated using wavelet analysis under the voltage of 3.5 V. The continuous wavelet analysis results are shown in Figure 19. Obvious high-frequency components are observed during the activation process (period A) and the operation process (period B). The amplitudes of the high-frequency components in periods A and B are roughly equal. The frequency corresponding to the amplitude of 1 is taken as the high frequency in the current during the corresponding time, and the current signal contains a 3 kHz component during the activation process and a 1.7 kHz component during the pick-up process. In the original static simulation model, the properties of soft magnetic materials are modified into BH curves at 3 kHz and 1.7 kHz, respectively, and the corresponding static data are obtained through simulations. The simulation model of the dynamic characteristics is shown in Figure 20.

As shown in Figure 21, the simulation process includes five stages. The first stage is the section during which the coil is not activated and the simulation model is in the initial state. The second stage is the beginning of the activation process. At this point, the coil current contains a 3 kHz high-frequency component. When the flux in the magnetic circuit changes from negative to positive, the high-frequency component of the coil current is very small, corresponding to simulation stage 3. When the coil current is maximized, the armature starts to rotate. In this process, the coil current contains a 1.7 kHz high-frequency component, corresponding to simulation stage 4. At the end of the pull-in process, the rotation angle is maximized. At this time, the high-frequency component in the coil current is very small, corresponding to simulation stage 5.

Under the coil voltage of 3.5 V, the comparison between the simulation and test results is as shown in Figure 22.

## 5. Results and Discussion

### 5.1. Replacement Design and Schemes

The bonding method of amorphous or nanocrystalline strips only allows simple shapes for bulk materials. Therefore, simple shapes are cut into corresponding parts. Then, the bulk materials with corresponding shapes are customized and fixed in the cut notch to complete the material replacement process.

Material replacements are applicable to the core, both ends of the armature, and parts of each yoke close to the armature of the electromagnetic system. The replacement methods are shown in Figure 23.

The design schemes are presented in Table 2.

### 5.2. Prototype Test and Simulation Results

The method described in Section 4 is used to simulate the dynamic characteristics of the model prepared with schemes A to F. The corresponding static data are input into the simulation circuit to obtain the corresponding dynamic simulation results, as shown in Figure 24 and Figure 25. Meanwhile, the experimental circuit is used to test the coil current under each replacement scheme and compare the tested results under different replacement schemes, as shown in Figure 26 and Figure 27. The tested operation time under all schemes is given in Table 3.

The above results show that, although schemes B/E yield the largest dynamic performance improvements in the simulation, the measured results of B/E are worse than those of schemes A/D due to the air gap between the yoke (or armature) and replacement part in the assembly process. The non-working air gap generated by material replacement increases the values of the coil current drop points under schemes B, C, E, and F, i.e., a larger current is needed to drive the armature to rotate. In addition, the static electromagnetic torque decreases, and the operation time increases due to the non-working air gap.

### 5.3. Full-Size Relay Test and Results

According to the above design schemes and test results, the use of amorphous and nanocrystalline materials can improve the dynamic characteristics of the relay to a certain extent. Considering the processing difficulty and optimization effect, design scheme A, which involves replacing the core with 1K107B, can be selected for the further processing and testing of the full-size relay (scheme D is more difficult to implement than scheme A in actual processing, and the amorphous core is more brittle than the nanocrystalline one).

Figure 28 shows the full-size relay with the 1K107B core (its size is shown in Figure 29), and Figure 30 shows the current curves of the relays before and after replacement in the operation process under 28 V. The electrical parameters of the relay with replacement and the original relay are tested, and the test results are as shown in Table 4.

According to the test results, the action process of the relay with replacement has a relatively obvious change compared with the original relay, and the operation time is still reduced by 30.8% when the pick-up voltage is increased by 2 V compared with the original one, which is also confirmed by Figure 29. After the relay with replacement is activated, the coil current rises rapidly. At this point, the magnetic field in the electromagnetic system builds up and reaches saturation more quickly. Compared with DT4C, the nanocrystalline 1K107B has more advantageous magnetic properties, maintaining relatively high permeability under high-frequency conditions, and the operation time is still greatly shortened when its saturation magnetic induction strength is inferior (its saturation magnetic induction strength is only approximately 1.25 T, while that of DT4C exceeds 1.8 T). This is sufficient to explain the effect of the high-frequency current on the armature’s operation.

## 6. Conclusions

This study takes the balanced force electromagnetic relay as an example and selects suitable amorphous and nanocrystalline soft magnetic materials based on a wavelet analysis. The effects of using the amorphous 1K101 and nanocrystalline 1K107B soft magnetic materials instead of pure iron for the core, armature, and yoke of the original scaled-up relay model on the dynamic performance of the electromagnetic relay were studied via prototype tests and simulation calculations. The experiment showed that amorphous and nanocrystalline magnetic materials can improve the dynamic characteristics of a relay as it operates. Then, the best scheme was selected to prepare and test a full-size relay. The test results show that the dynamic characteristics of the relay were greatly improved, while the operation time was reduced by 30.8%. The reason is speculated to be the significant effect of the high-frequency magnetic properties of the material on the process in which the relay operates, but further research is needed to study the application of new materials in electromagnetic relays. In the future, we will comprehensively analyze the needs of balanced-force-type sealed electromagnetic relays and the performance of nanocrystalline materials, and we will develop corresponding magnetic materials so as to obtain a complete theoretical analysis and verification and a greater performance improvement.

## Figures and Tables

**Figure 1 micromachines-15-00368-f001:**
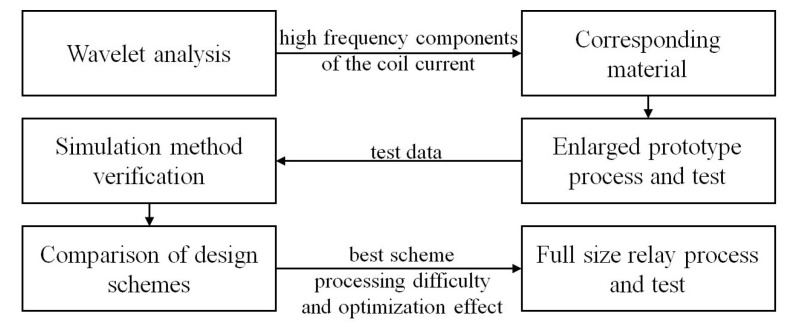
Experimental procedure.

**Figure 2 micromachines-15-00368-f002:**
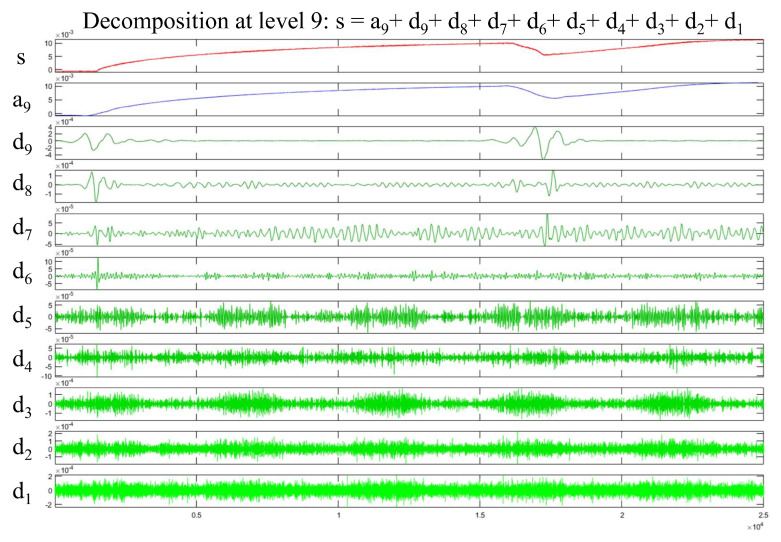
Wavelet analysis results obtained with sym3 function.

**Figure 3 micromachines-15-00368-f003:**
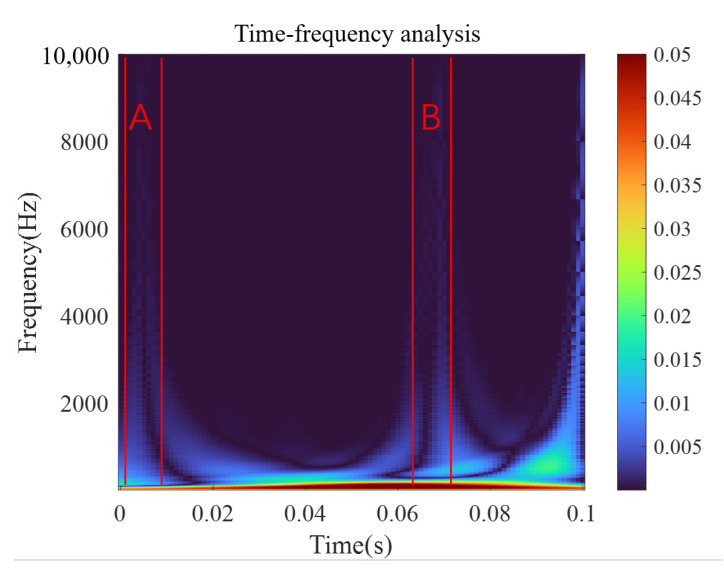
Wavelet time–frequency analysis results.

**Figure 4 micromachines-15-00368-f004:**
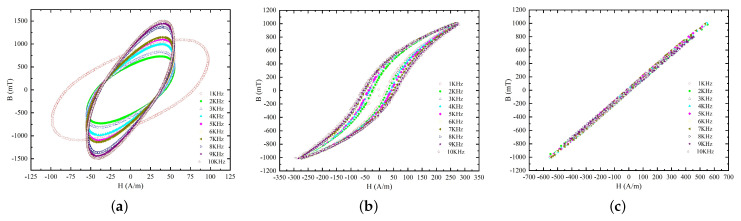
High-frequency BH curves of different materials (1–10 kHz): (**a**) DT4C. (**b**) 1K101. (**c**) 1K107B.

**Figure 5 micromachines-15-00368-f005:**
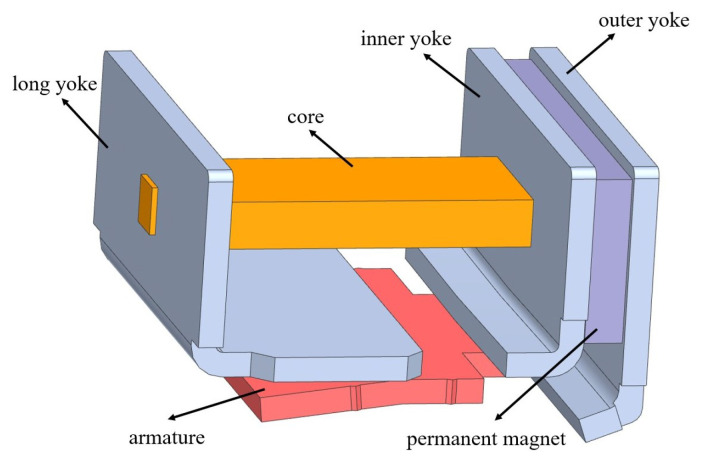
The original model.

**Figure 6 micromachines-15-00368-f006:**
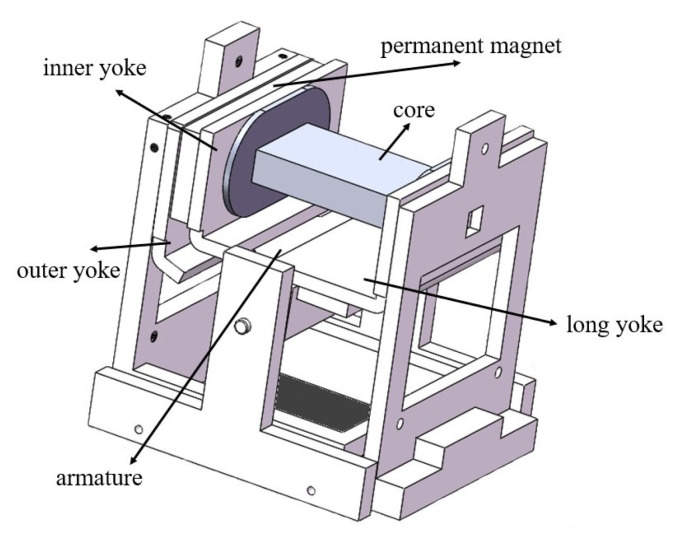
The scaled-up model.

**Figure 7 micromachines-15-00368-f007:**
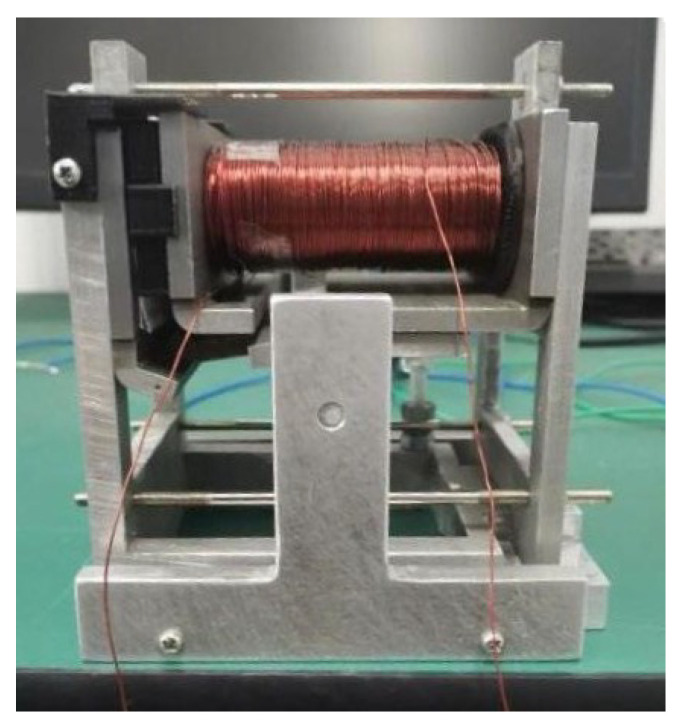
The scaled-up relay prototype.

**Figure 8 micromachines-15-00368-f008:**
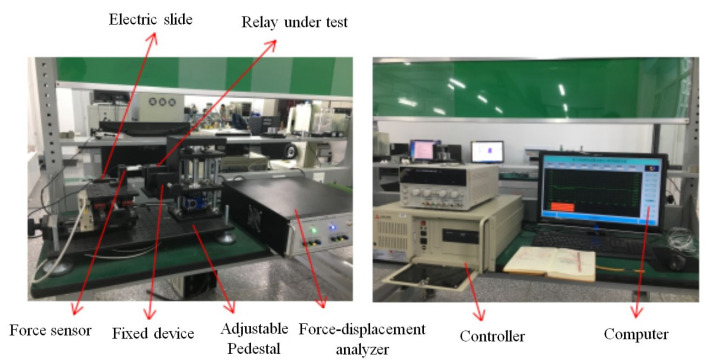
Static characteristic test equipment.

**Figure 9 micromachines-15-00368-f009:**
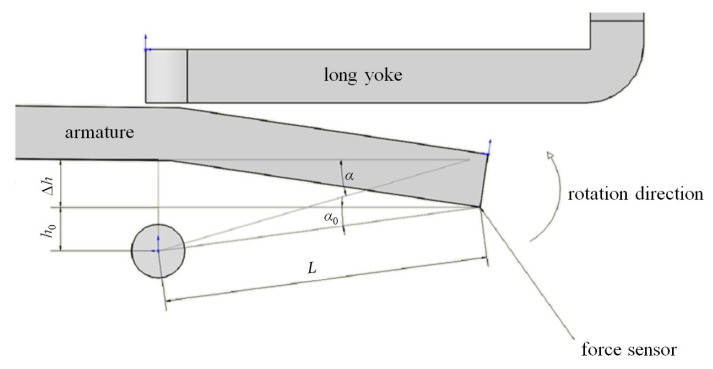
The conversion relationship between the displacement and the rotation angle.

**Figure 10 micromachines-15-00368-f010:**
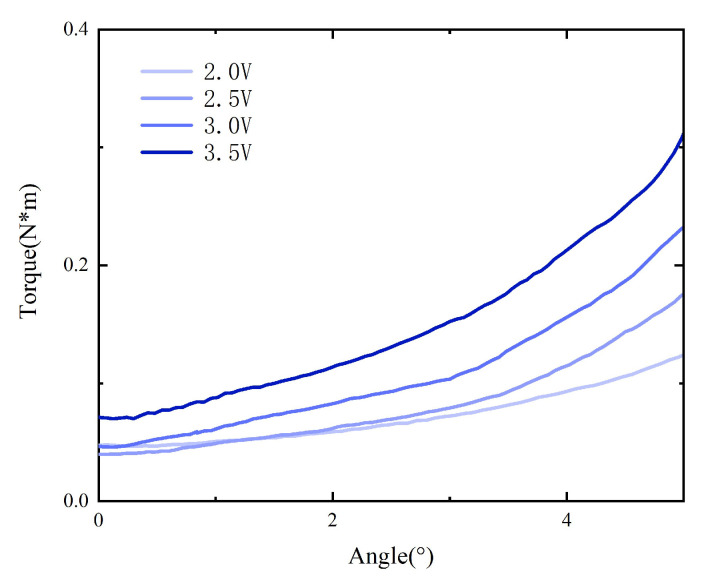
Curves (M−Δα) of the original model.

**Figure 11 micromachines-15-00368-f011:**
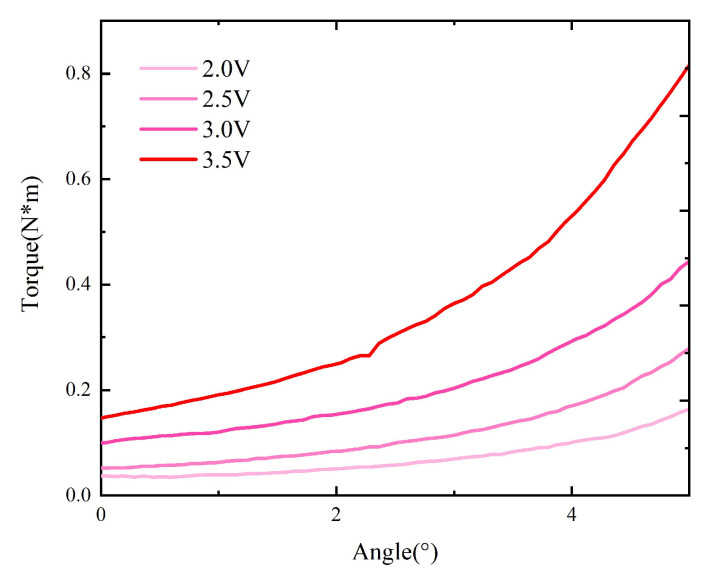
Curves (M−Δα) of the model with replacement.

**Figure 12 micromachines-15-00368-f012:**
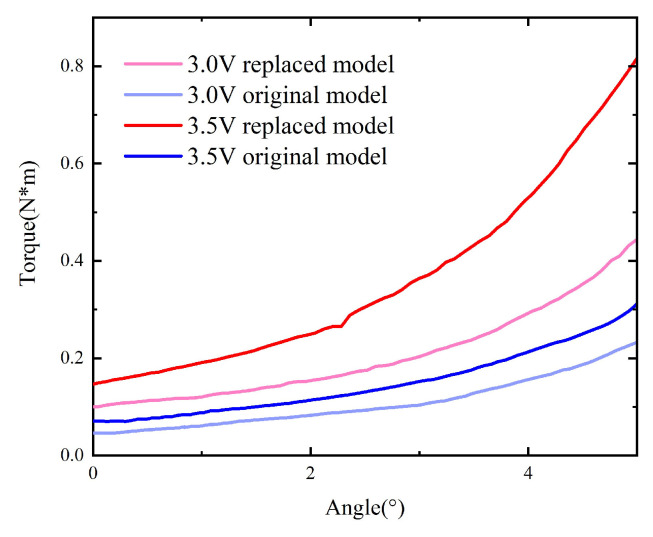
Comparison between the original model and the one with replacement.

**Figure 13 micromachines-15-00368-f013:**
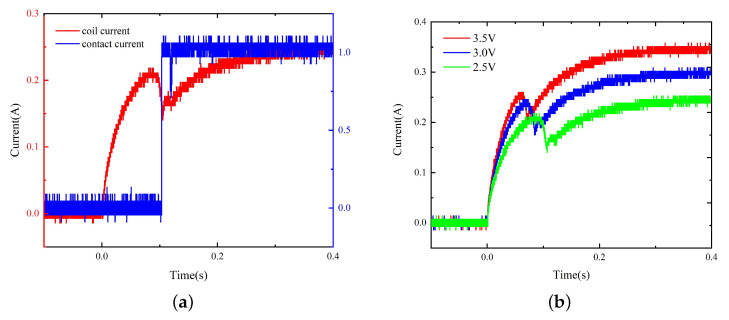
(**a**) The coil current and contact current of the original model under 2.5 V. (**b**) The coil current curves of the original model under 2.5 V to 3.5 V.

**Figure 14 micromachines-15-00368-f014:**
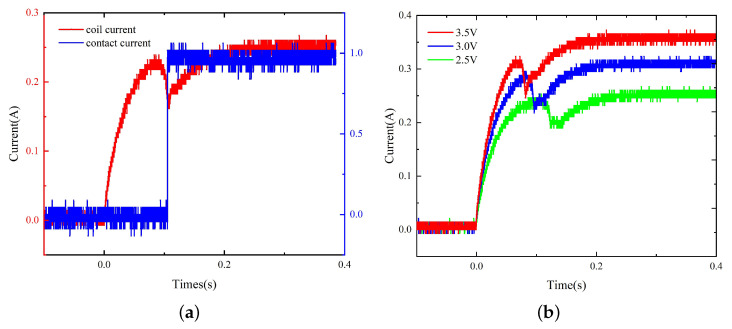
(**a**) The coil current and contact current of the model with replacement under 2.5 V. (**b**) The coil current curves of the model with replacement under 2.5 V to 3.5 V.

**Figure 15 micromachines-15-00368-f015:**
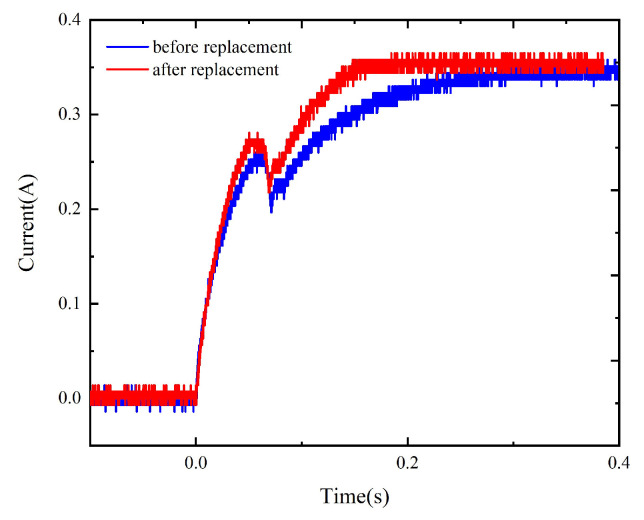
The dynamic characteristics of the model before and after material replacement.

**Figure 16 micromachines-15-00368-f016:**
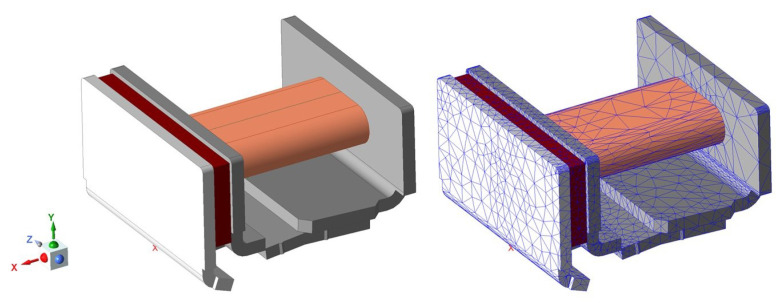
The finite element model and meshed model.

**Figure 17 micromachines-15-00368-f017:**
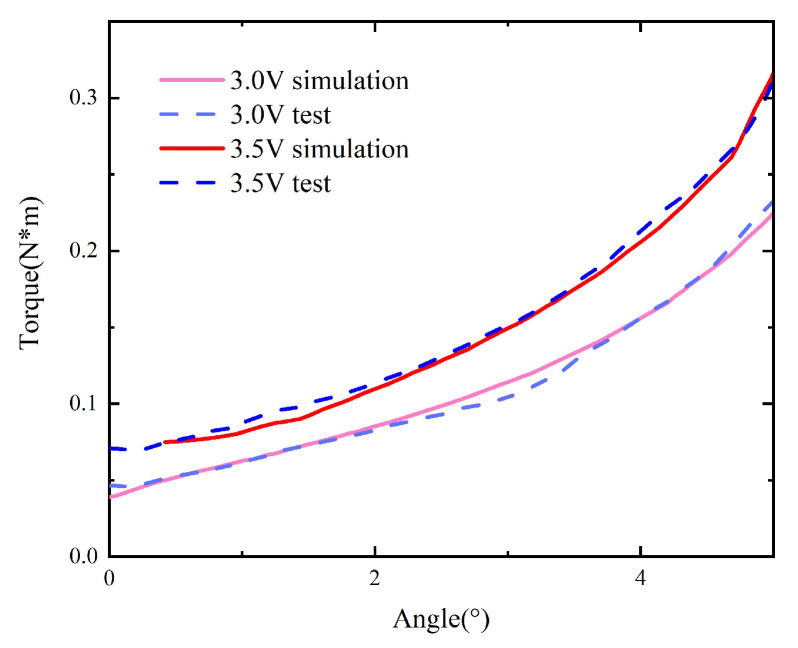
The simulation and test results of the original model.

**Figure 18 micromachines-15-00368-f018:**
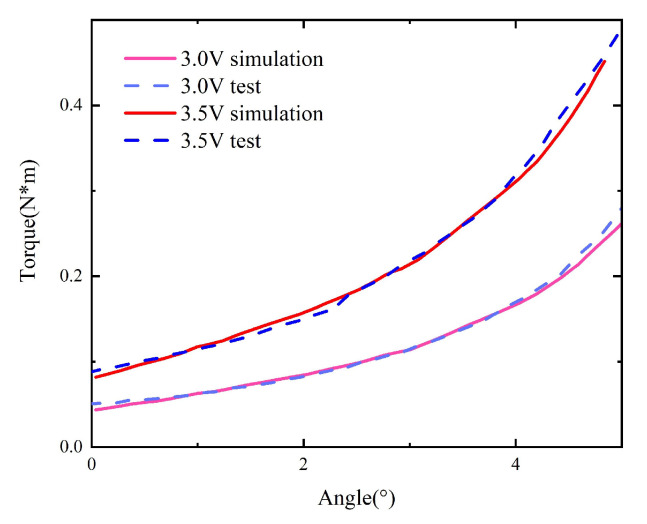
The simulation and measurement results of the model with replacement.

**Figure 19 micromachines-15-00368-f019:**
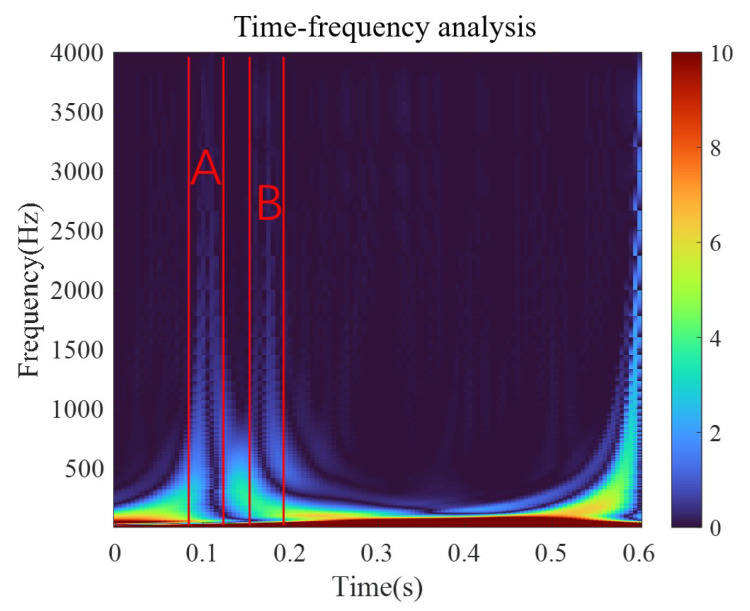
Wavelet analysis results of the original model under 3.5 V.

**Figure 20 micromachines-15-00368-f020:**
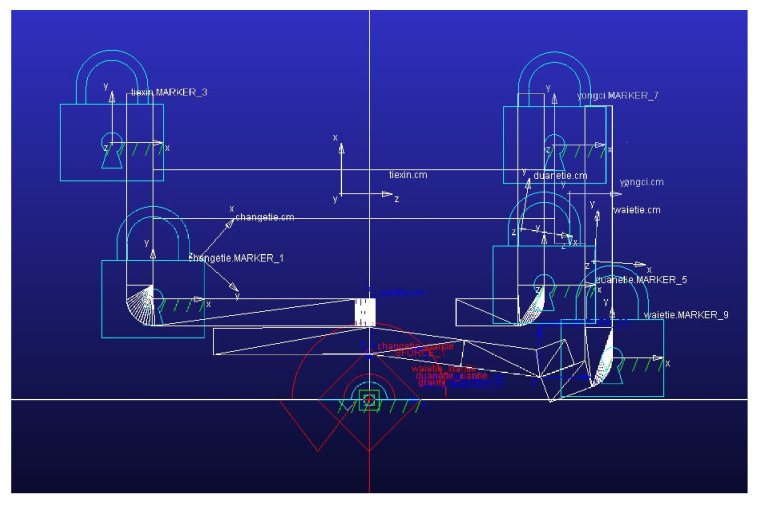
The simulation model of the dynamic characteristics.

**Figure 21 micromachines-15-00368-f021:**
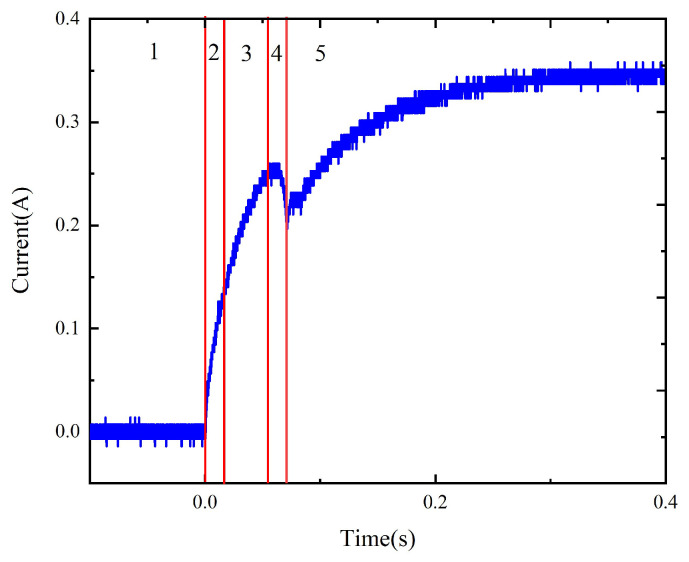
Five stages of the simulation process.

**Figure 22 micromachines-15-00368-f022:**
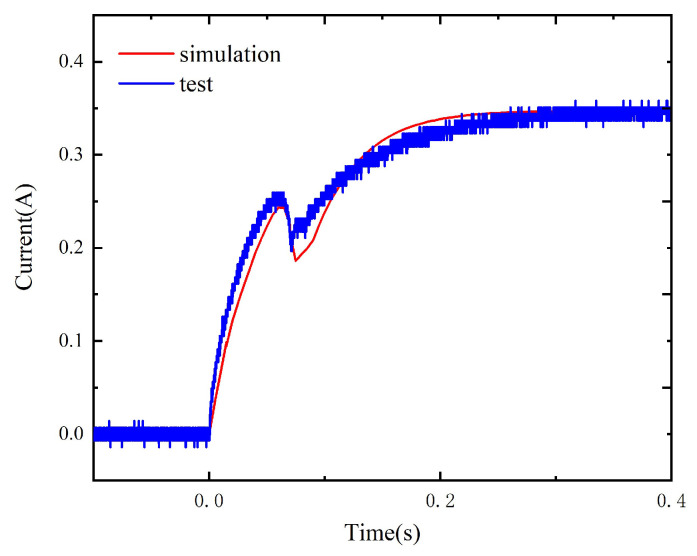
The simulation and test results.

**Figure 23 micromachines-15-00368-f023:**
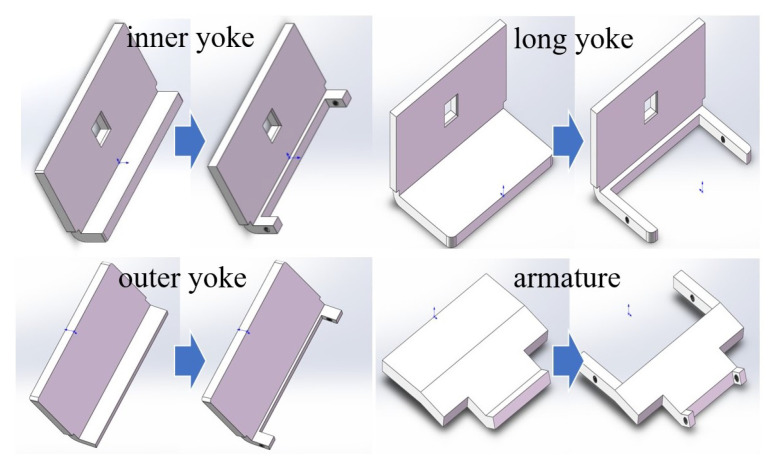
The material replacement methods.

**Figure 24 micromachines-15-00368-f024:**
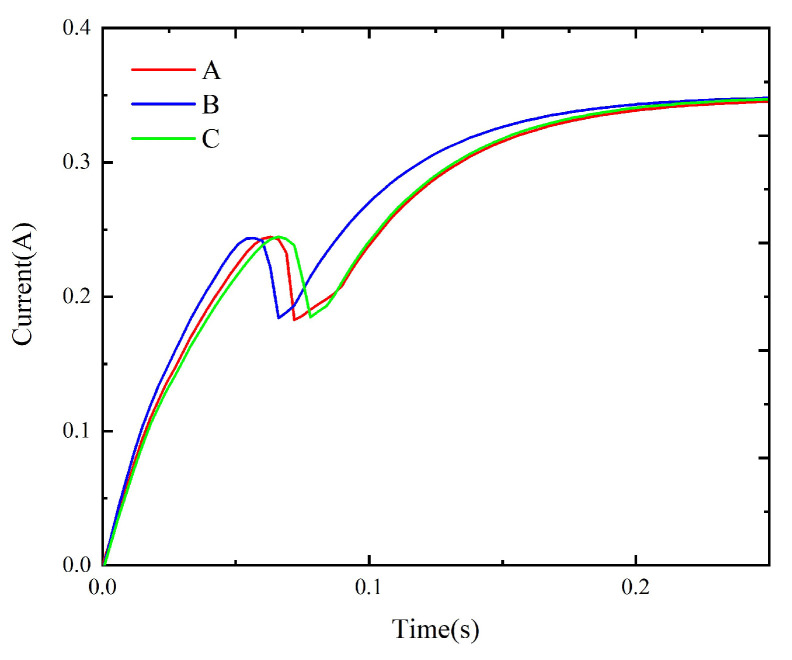
The simulation results under schemes A, B, and C.

**Figure 25 micromachines-15-00368-f025:**
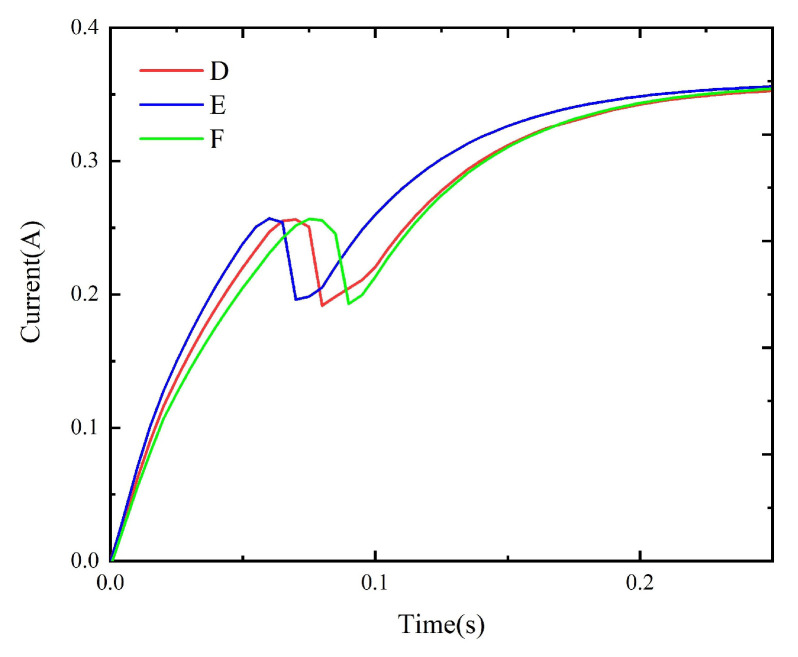
The simulation results under schemes D, E, and F.

**Figure 26 micromachines-15-00368-f026:**
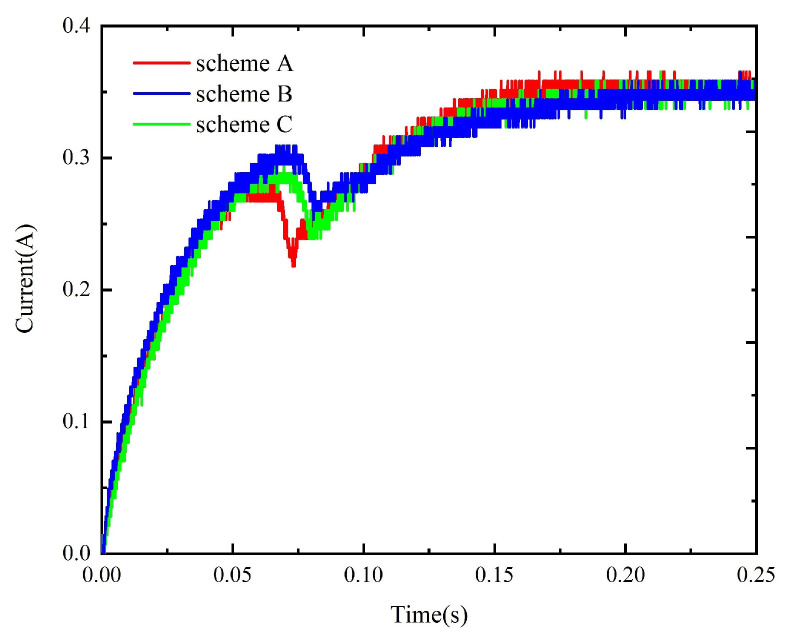
The test results under schemes A, B, and C.

**Figure 27 micromachines-15-00368-f027:**
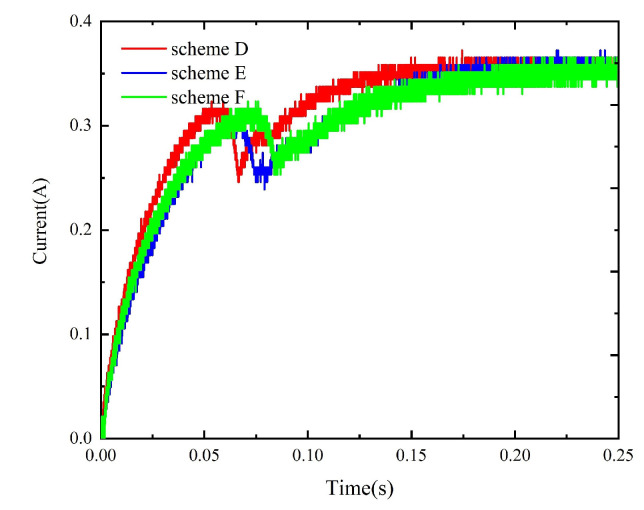
The test results under schemes D, E, and F.

**Figure 28 micromachines-15-00368-f028:**
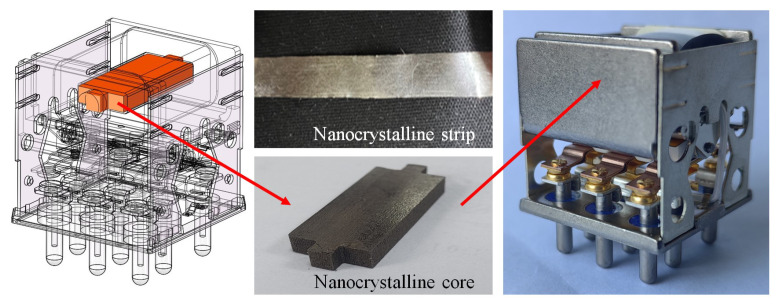
The full-size relay with the 1K107B core.

**Figure 29 micromachines-15-00368-f029:**
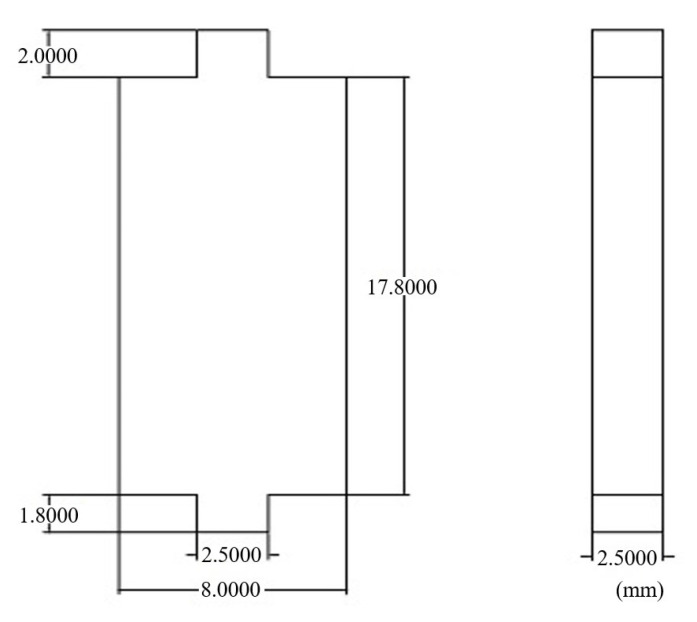
The size of the core.

**Figure 30 micromachines-15-00368-f030:**
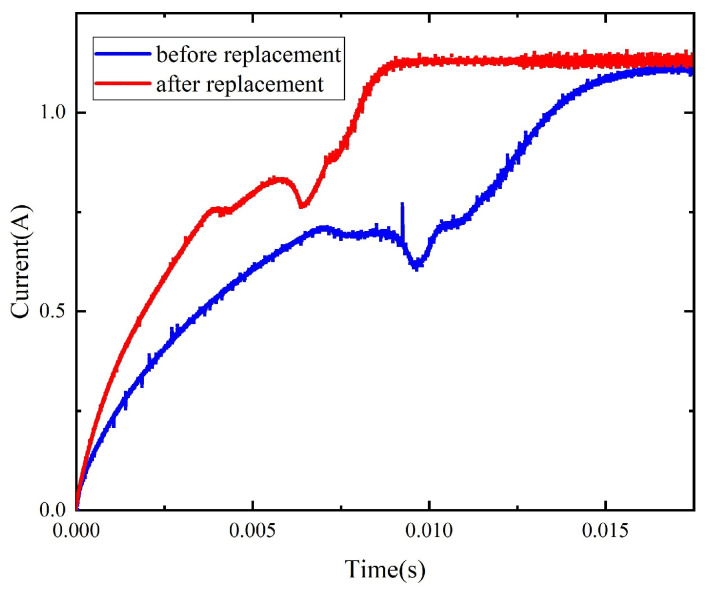
The coil current curves of the original relay and the relay with replacement under 28 V.

**Table 1 micromachines-15-00368-t001:** The operation time before and after material replacement under different voltages.

Coil Voltage/V	Original Model/ms	Model with Replacement/ms
2.5	107.8	100.7
3	80.9	74.8
3.5	71.4	65.7

**Table 2 micromachines-15-00368-t002:** The design schemes.

Number	Schemes
A	The core is replaced with nanocrystals.
B	The core, yoke, and armature are replaced with nanocrystals.
C	The yoke and armature are replaced with nanocrystals.
D	The core is replaced with an amorphous alloy.
E	The core, yoke, and armature are replaced with an amorphous alloy.
F	The yoke and armature are replaced with an amorphous alloy.

**Table 3 micromachines-15-00368-t003:** The test results under all schemes.

Scheme	Operation Time/ms		
	2.5 V	3.0 V	3.5 V
A	100.7	74.8	65.7
B	103.1	92.3	77.3
C	99.3	91.2	80.2
D	85.6	78.4	64.3
E	116.6	100.2	76.2
F	108.7	90.6	83.4
Original model	107.8	80.9	71.4

**Table 4 micromachines-15-00368-t004:** The electrical parameters of the original relay and the relay with replacement.

Number	Operation Time/ms	Pick-Up Voltage/V	Release Time/ms	Drop-Out Voltage/V
Relay with replacement	5.96	16.0	1.98	2.5
Original relay	8.61	14.0	2.45	3.0

## Data Availability

Data are contained within the article.

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
