# Peer review of "Application of Amorphous and Nanocrystalline Soft Magnetic Materials in Balanced-Force-Type Electromagnetic Relay"

_micromachines, 2024, doi:10.3390/mi15030368_

Round 1
Reviewer 1 Report
Comments and Suggestions for Authors
This paper addresses an interesting work on a type of electromagnetic relay. Nanocrystalline soft magnetic materials are used in the magnetic circuit of the magnetic relay to improve its dynamic characteristics. The principle of selecting nanocrystalline materials depends on the wavelet analysis of the high-frequency components of the coil current during the operation of the relay. A scale-up prototype is built and simulation method supports this research. Different application situations of six typical schemes using amorphous and nanocrystalline materials are proposed. The test results proved that this application attempt could improve the relay dynamic characteristics. This paper demonstrates a new application in such an electromagnetic system, which, to my knowledge, has never been attempted before.
There are some commons for authors to improve this paper:
1) What is the iron loss in the magnetic material of a relay electromagnetic system before and after using nanocrystalline materials? Please give a brief description.
2) It seems a bit strange to build a prototype at the beginning. Usually, simulations can be done after theoretical analysis. Prototypes can be created for testing during the validation phase. It would have been much clearer if the authors had provided some explanations in this paper.
3) Some grammatical errors could be improved, please check the whole article.
Reviewer 2 Report
Comments and Suggestions for Authors
This paper uses amorphous and nanocrystalline soft magnetic materials with good high-frequency magnetic properties in magnetic circuits.Six amorphous and nanocrystalline material alternatives are proposed based on the analysis and simulation results of high-frequency signals and dynamic characteristics, and real-size samples of the nanocrystalline alloy 1K107B core have been prepared as a result.
Overall, it is an informative paper with important technical details on the application of soft magnetic materials in relays. However, it also consists of some minor shortcomings that must be corrected before being considered for publication.
1. Figure 3 Frequency colour change is not obvious, it is recommended to change the colour chart, the second partial chart is not uniform in horizontal coordinate units.
2. Figure 19 Frequency colour change is not obvious, it is recommended that the colour table be replaced and the vertical coordinates of the local graph be replaced with English.
3. Replacement sizes for 1K107B material used not described in subsection 5.3.
4. The shortened operating time of the relay causes a transient load that has an effect on the stability of the motor, please add arguments as to whether it has an adverse effect on it.
Comments on the Quality of English LanguageThe Quality of English Language should be improved
Reviewer 3 Report
Comments and Suggestions for Authors
The micro sealed electromagnetic relays used in the aerospace field have high requirements for product performance and reliability. This type of relay uses traditional soft magnetic materials typically. The research work in this paper attempts to explore the application of nanocrystalline magnetic materials in aerospace electromagnetic relay, supported by theoretical analysis. No previous scholars have studied this research direction. Starting from the transient process of relays, this article explores the idea of theoretically matching the high-frequency characteristics of nanocrystalline materials to meet the needs of improving transient performance. Prototype has been established, achieving the effect of improving dynamic characteristics and verifying the correctness of this analysis and design method.
The theoretical analysis of the article is relatively complete, and the design and experimental process are reasonable. Some comments and improve suggestions show below:
1 The FEM analysis of magnetic materials containing nanocrystals poses certain challenges. Please provide more details about mesh process, calculation time consuming details, and model size in the article;
2 In the beginning of the theoretical and FEM analysis, authors established a physical model of electromagnetic mechanism, whether it is part of the simulation analysis process, and why this model should be built in the early stage, please give more details;
3 The nanocrystalline magnetic materials used in this research are standard type of nanocrystalline magnetic materials. Based on the experience of similar problems in the field of motors (such as iron loss, noise, vibration, assembly technics, etc.), it is recommended that the author analyze more accurate and comprehensive requirements for nanocrystalline magnetic material for such sealed electromagnetic relays in near future research, and prepare corresponding magnetic materials in a targeted manner, to verify your theoretical analysis and performance improvement methods much better.
Hope to read more research papers in this interesting field from authors.
Comments on the Quality of English LanguageThe overall English expression is easy to understand, but there still are some grammar and expression errors in details. Please carefully check the entire article.
